# An Iterative Self-Learning Framework for Medical Domain Generalization

**Zhenbang Wu** [1]    **Huaxiu Yao** [2]    **David M Liebovitz** [3]    **Jimeng Sun** [1]

[1] University of Illinois Urbana-Champaign, {zw12, jimeng}@illinois.edu
[2] University of North Carolina at Chapel Hill, huaxiu@cs.unc.edu
[3] Northwestern University, david.liebovitz@nm.org

## Abstract

Deep learning models have been widely used to assist doctors with clinical decision-making. However, these models often encounter a significant performance drop when applied to data that differs from the distribution they were trained on. This challenge is known as the domain shift problem. Existing domain generalization algorithms attempt to address this problem by assuming the availability of domain IDs and training a single model to handle all domains. However, in healthcare settings, patients can be classified into numerous latent domains, where the actual domain categorizations are unknown. Furthermore, each patient domain exhibits distinct clinical characteristics, making it sub-optimal to train a single model for all domains. To overcome these limitations, we propose SLDG, a self-learning framework that iteratively discovers decoupled domains and trains personalized classifiers for each decoupled domain. We evaluate the generalizability of SLDG across spatial and temporal data distribution shifts on two real-world public EHR datasets: eICU and MIMIC-IV. Our results show that SLDG achieves up to 11% improvement in the AUPRC score over the best baseline.

## 1 Introduction

Deep learning techniques have been increasingly popular in clinical predictive modeling with electronic health records (EHRs) [12, 11, 47, 58, 2]. However, these models typically assume that the training (source) data and testing (target) data share the same underlying data distribution (i.e., domain). This assumption can become problematic when models are applied to new domains, such as data from different hospitals or future time points [17, 59, 20, 37]. In these situations, domain shifts caused by variations in patient cohorts, clinical standards, and terminology adoption can significantly degrade the model's performance.

This paper aims to develop a clinical predictive model on the source data that effectively handles potential domain shifts when applied to the target data. Domain generalization (DG) [7] methods have been widely utilized to address such problems, including techniques like domain alignment [32, 24, 25, 45, 31, 51, 62], meta-learning [23, 27, 26, 5, 22, 29], and ensemble learning [9, 42, 43, 61]. However, when applied in healthcare settings, these methods encounter the following limitations:

- **Reliance on domain IDs.** Most DG methods depend on the presence of domain IDs, which indicate the domain to which each sample belongs, to guide the model training [24, 16, 4, 9, 42]. However, as shown in Fig. 1, patients can be divided into numerous latent domains based on features such as age, medical history, treatment, and symptoms. The actual categorization of these latent domains can be difficult to obtain and vary across different tasks [1, 49]. Consequently, existing DG methods often resort to broad domain categorizations, such as hospital or timestamp, which provide limited information [59, 17].

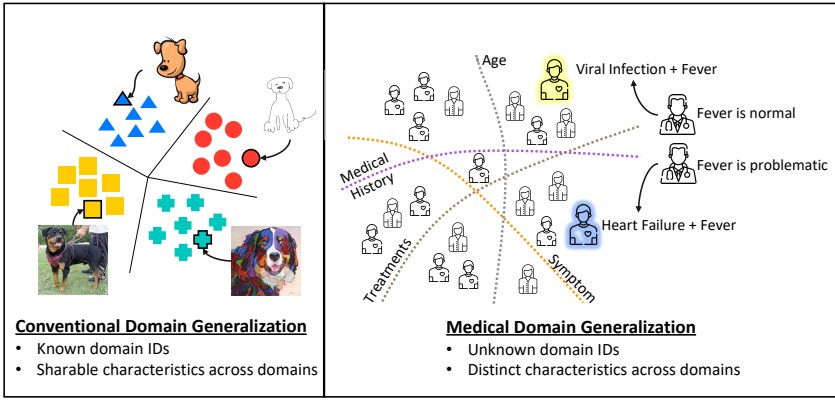

Figure 1: Conventional domain generalization methods typically rely on domain IDs and shared characteristics across domains to train a single generalized model. However, in the medical field, patients can be classified into numerous latent domains that are not directly observable. Additionally, each patient domain exhibits unique clinical characteristics, making it sub-optimal to train a single model for all domains.

- **Attempt to train a single model.** While some recent DG methods have attempted to alleviate the reliance on domain labels [62, 29], they try to train a single model that generalizes across all domains. However, patients from different domains possess distinct characteristics and require different treatment approaches [2, 58]. For example, as shown in Fig. 1, fever is considered a normal symptom for patients with viral infections as it helps stimulate the immune system. On the other hand, it can be a bad signal for patients with cardiovascular disease, leading to complications. Thus, training a single model for all domains is challenging and can lead to sub-optimal performance.

To overcome these limitations, we propose SLDG, a self-learning framework for domain generalization that iteratively discovers decoupled domains and trains customized classifiers for each discovered domain. Specifically, SLDG consists of the following iterative steps:

- **Decoupled domain discovery.** While domain labels are not initially available, we posit that they can be recovered by clustering the learned latent representations. However, identifying all fine-grained domains across various clinical features (e.g., demographics, diagnosis, and treatments) can be challenging. Instead, we propose to decouple these clinical features and discover the clusters separately for each type of feature. To achieve this, we maintain a distinct latent space for each type of features using a feature-specific patient encoder. Within each latent space, we perform hierarchical clustering independently to discover the domain categorizations. By adopting this approach, we effectively reduce the number of domains from exponential to linear to the number of feature types.

- **Domain-specific model customization.** To account for the unique characteristics of patients in different domains, our approach involves training customized classifiers for each domain. To ensure parameter efficiency, we extract domain representations from the learned clusters and utilize them to parameterize the domain-specific classifiers. For a given patient, we determine the closest domain by comparing the patient's representations with the domain representations, and subsequently select the corresponding classifier for accurate inference.

To assess the generalizability of SLDG across spatial and temporal shifts, we conduct experiments on two publicly available EHR datasets: eICU [38] and MIMIC-IV [18]. Our results demonstrate that SLDG outperforms the best baseline by up to 11% in terms of AUPRC score. We also conduct detailed analyses and ablation studies to investigate the factors contributing to the performance gain achieved by SLDG.

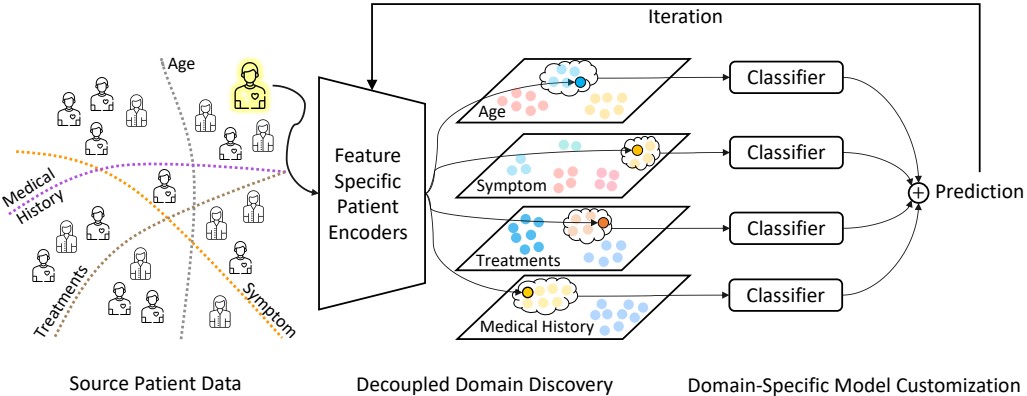

Figure 2: An illustration of the `SLDG` framework. The feature-specific patient encoder maps each patient into multiple latent spaces, with each space capturing patient characteristics from a specific perspective. Next, `SLDG` iteratively performs decoupled clustering to identify latent domains and learns domain-specific classifiers customized for each domain.

## 2   Preliminaries

In EHR data, a patient's hospital visit is represented by a sequence of events, denoted as $x = [e_1, e_2, \ldots, e_m]$, where $m$ is the total number of events in the visit. Each event $e$ characterizes features of a certain type $t$, such as diagnosis, prescription, and lab tests. This mapping is denoted by a function $T(e) : \mathcal{E} \to \mathcal{T}$, where $\mathcal{E}$ and $\mathcal{T}$ denote the sets of all events and types, respectively. For example, a patient visit can be [Acute embolism (I82. 40), Atrial fibrillation (I48.91), Ultrasound (76700), CT scan (G0296), ECG (93042), Heparin IV (5224)]. And each event corresponds to a specific type (e.g., diagnosis, procedure, medication). The main objective of clinical predictive modeling is to predict the occurrence of future events, such as 15-day hospital readmission and 90-day mortality, denoted as $y \in \{+, -\}$, based on the patient's current visit $x$.

Existing clinical prediction works typically train a model $f_\phi(\cdot)$ with parameter $\phi$ by minimizing a loss function $l(\cdot)$ on source training data sampled from distribution $P_{tr}$, as in Eq. (1),

$$\arg\min_\phi \mathbb{E}_{(x,y) \sim P_{tr}}[l(f_\phi(x), y)], \tag{1}$$

with the hope that the trained model can perform well on the target test data distributed according to $P_{te}$. However, in real-world settings, the source and target distributions can differ due to spatial and temporal shifts, i.e., $P_{tr} \neq P_{te}$. Consequently, the model trained on source data may experience a drop in performance when applied to the target data.

## 3   The `SLDG` Approach

In this paper, our goal is to train a model $f_\phi(\cdot)$ on source data $P_{tr}$ that can generalize to target data $P_{te}$ despite potential domain shifts. Existing DG algorithms face limitations due to their reliance on domain IDs and attempts to train a single model for all domains. To overcome these limitations, we propose to iteratively discover latent domains and train customized classifiers for each domain. However, we face the challenge of dealing with a large number of latent domains, which not only makes domain discovery difficult but also results in an exponential increase in the number of model parameters with respect to the number of feature types. In the following sections, we will describe how our method SLDG addresses this challenge through decoupled domain discovery and domain-specific model customization. Additionally, we will introduce the training and inference strategy. Fig. 2 illustrates the `SLDG` framework.

### 3.1   Decoupled Domain Discovery

Although domain labels are not initially available, we hypothesize that the domain information is encoded in the learned latent representations and can be recovered with the clustering technique. How-

ever, patients can be categorized into thousands of latent domains, determined by various features such as age, medical history, treatment, and symptoms. For instance, a patient can fall into the fine-grained domain of *older male patients with a history of smoking and a diagnosis of type 2 diabetes*. Identifying all such fine-grained domains can be challenging, as clustering methods may either overlook smaller domains or result in an excessive number of domains that would inflate the number of parameters in subsequent steps. To address this, we propose decoupling these clinical features and independently discovering clusters for each feature type. For example, the patient above can simultaneously belong to the decoupled domains of *older*, *male*, *history of smoking*, and *diagnosis of type 2 diabetes*. This approach effectively reduces the number of domains from exponential to linear with respect to the number of feature types.

Concretely, we maintain a distinct latent space for each type of feature. When given an input patient visit $x$, SLDG maps it to the latent space corresponding to the feature type $t \in \mathcal{T}$ using a feature-specific patient encoder $E_t(\cdot)$, as in Eq. (2),

$$\mathbf{h}_t := E_t(x), \quad \mathbf{h}_t \in \mathbb{R}^h, \tag{2}$$

where $h$ denotes the hidden dimension. Next, within each latent space of type $t$, SLDG gathers all patient representations $\{\mathbf{h}_t^{(i)}\}_{i=1}^{N_{tr}}$, where $N_{tr}$ is the number of source training data, and performs clustering to discover the domain categorizations, as in Eq. (3),

$$\mathbf{M}_t := \text{Cluster}(\{\mathbf{h}_t^{(i)}\}_{i=1}^{N_{tr}}), \quad \mathbf{M}_t \in \{0, 1\}^{N_{tr} \times K_t}, \tag{3}$$

where $K_t$ represents the number of discovered domains in the latent space of type $t$. $\mathbf{M}_t$ denotes the learned domain assignment, where $\mathbf{M}_t[i, k]$ is equal to one if and only if (i.f.f.) the patient $x^{(i)}$ is assigned to the $k$-th domain. We will describe this procedure in detail in the following.

**Feature-Specific Patient Encoding.** This module is responsible for mapping each patient into multiple latent spaces, each capturing the patient's health status of a specific feature type. This enables subsequent modules to decouple the representations of different feature types. For a patient's hospital visit $x$ with a list of events $[e_1, \ldots, e_m]$, SLDG computes the contextualized representation for each event by applying the embedding function $E(\cdot)$, as in Eq. (4),

$$[\mathbf{e}_1, \ldots, \mathbf{e}_m] = E([e_1, \ldots, e_m]), \quad \mathbf{e}_j \in \mathbb{R}^h, \tag{4}$$

where $\mathbf{e}_j$ is the contextualized representation for event $e_j$ with dimension $h$. We model $E(\cdot)$ using a three-layer Transformer [48] framework. To ensure that there are no unseen events in the target data, we initialize the event embedding look-up table with ClinicalBERT [3] embeddings of the event name and then project it down to our hidden dimension of size $h$. The embedding look-up table is fixed during training.

Next, SLDG aggregates the contextualized event representations $[\mathbf{e}_1, \ldots, \mathbf{e}_m]$ based on their types, such as family history, diagnosis, and treatments. For each type $t \in \mathcal{T}$, the type-specific representation $\mathbf{h}_t$ is computed by averaging the representations of all events of that type, as in Eq. (5),

$$\mathbf{h}_t = \text{Average}(\{\mathbf{e}_j \mid T(e_j) = t\}_{j=1}^m), \quad \mathbf{h}_t \in \mathbb{R}^h, \tag{5}$$

where $T(e_i)$ indicates the type of event $e_i$. If no events belong to a certain type, the pooled sequence representation is used as a substitute. Consequently, each patient's hospital visit is represented by a set of vectors $\{\mathbf{h}_t\}_{t \in \mathcal{T}}$, with each vector capturing the patient's health status from a specific type of events. These decoupled patient representations are then utilized to perform per-feature-type domain clustering, described next.

**Hierarchical Domain Clustering.** This module is responsible for clustering patient representations in each latent space to discover latent domains, enabling subsequent modules to customize the classifier for each domain. In the previous step, we obtain a set of patient representations $\{\mathbf{h}_t^{(i)}\}_{i=1}^{N_{tr}}$ for each latent space of type $t$. To perform clustering, standard clustering techniques such as k-Means and Gaussian Mixture Model (GMM) require specifying the number of clusters, which is less ideal as the number of clusters can be difficult to choose and may vary across latent spaces. Inspired by GEORGE [46], we adopt a fully automated hierarchical clustering technique by monitoring the Silhouette score [40].

Specifically, in each latent space, `SLDG` first applies UMAP [30] for dimensionality reduction. Then, it runs k-Means with $k \in \{2, \ldots, 10\}$ to identify the optimal number of clusters based on the highest Silhouette score. Subsequently, `SLDG` further split each cluster into five sub-clusters. However, only sub-clusters surpassing the Silhouette score of the original cluster and containing at least 500 patients are retained. The final number of clusters in the latent space of type $t$ is denoted as $K_t$. The cluster assignment is represented by a binary matrix $\mathbf{M}_t$ of size $N_{tr} \times K_t$, where $\mathbf{M}_t[i, k]$ is set to one i.f.f. the patient $x^{(i)}$ is assigned to the $k$-th cluster. This automated approach allows us to effectively select the number of clusters in each latent space, balancing between discovering overly coarse or fine-grained clusters.

## 3.2 Domain-Specific Model Customization

To accommodate the unique characteristics of patients in different domains, we propose to train customized classifiers for each decoupled domain. Given an input patient visit $x$ and its multi-vector representations $\{\mathbf{h}_t\}_{t \in \mathcal{T}}$, `SLDG` computes the predicted probability $o$ of a specific event occurring by employing a weighted combination of domain-specific classifiers in each latent space $t \in \mathcal{T}$, as in Eq. (6),

$$o := \frac{1}{|\mathcal{T}|} \sum_{t \in \mathcal{T}} \sum_{k=1}^{K_t} \underbrace{G_{t,k}(\mathbf{h}_t)}_{\text{gate}} \cdot \underbrace{C_{t,k}(\mathbf{h}_t)}_{\text{classifier}}, \quad o \in \mathbb{R}, \tag{6}$$

where $C_{t,k}(\cdot)$ refers to the customized classifier for the discovered domain $k$ in the latent space of type $t$, while $G_{t,k}(\cdot)$ corresponds to the gating function. In the following, we will elaborate on how `SLDG` leverages the clustering results to efficiently parameterize the domain-specific classifier and effectively determine the gating weights.

To efficiently parameterize the domain-specific classifier $C_{t,k}(\cdot)$ for the $k$-th discovered domain in the latent space of type $t$, we define two learnable weight vectors of size $h$: $\mathbf{w}_{t,k}^+$ and $\mathbf{w}_{t,k}^-$, which represent the prototypes of the positive and negative classes, respectively. The predicted probability of a specific event occurring is computed based on the relative distance between the patient representation $\mathbf{h}_t$ and the positive and negative prototypical weights, as in Eq. (7),

$$C_{t,k}(\mathbf{h}_t) = \frac{\exp(-d(\mathbf{w}_{t,k}^+, \mathbf{h}_t))}{\sum_{* \in \{+,-\}} \exp(-d(\mathbf{w}_{t,k}^*, \mathbf{h}_t))}, \quad C_{t,k}(\mathbf{h}_t) \in \mathbb{R}, \tag{7}$$

where $d(\cdot, \cdot)$ is the Euclidean distance. To facilitate efficient learning, we initialize the two prototypical weight vectors $\mathbf{w}_{t,k}^+$ and $\mathbf{w}_{t,k}^-$, with the average representations of patients from the corresponding classes, as in Eq. (8),

$$\text{Init}(\mathbf{w}_{t,k}^*) = \text{Average}(\{\mathbf{h}_t^{(i)} \mid (M_t[i, k] = 1) \wedge (y^{(i)} = *)\}_{i=1}^{N_{tr}}), \quad * \in \{+, -\}, \tag{8}$$

where $M_t[i, k] = 1$ includes only patients assigned to the $k$-th domain in the latent space of type $t$.

We adopt a similar approach to the gating function $G_t(\cdot)$. For each discovered domain $k$ in the latent space of type $t$, we introduce a learnable prototypical weight vector $\mathbf{w}_{t,k} \in \mathbb{R}^h$. The gating weights are determined based on the distance between the patient representation $\mathbf{h}_t$ and the corresponding prototypical weights, as in Eq. (9),

$$G_t(\mathbf{h}_t) = \text{Softmax}(\{-d(\mathbf{w}_{t,k}, \mathbf{h}_t)\}_{k=1}^{K_t}), \quad G_t(\mathbf{h}_t) \in \mathbb{R}^{K_t}, \tag{9}$$

where the prototypical weight vectors $\{\mathbf{w}_{t,k}\}_{k=1}^{K_t}$ are initialized as the average representations of patients in that domain, as in Eq. (10),

$$\text{Init}(\mathbf{w}_{t,k}) = \text{Average}(\{\mathbf{h}_t^{(i)} \mid M_t[i, k] = 1\}_{i=1}^{N_{tr}}), \quad k = 1, \ldots, K_t. \tag{10}$$

## 3.3 Training and Inference

To train `SLDG`, we begin by utilizing a pre-trained patient encoder [1] for decoupled domain discovery. Then, we iteratively update the model weights and re-generate the clusters every 20 epochs. In each

---

[1] In practice, we pre-train the patient encoder on the same clinical predictive task for 40 epochs. This means that `SLDG` undergoes a total of 100 epochs of training, which is consistent with other baselines.

iteration, we re-initialize the classifier and gating parameters. This iterative process is repeated three times to enhance the model's performance. During training, we minimize the binary cross-entropy loss. For inferencing, given a target patient visit, SLDG first maps it to multiple decoupled latent spaces with the feature-specific patient encoders $E_t(\cdot)$. Subsequently, in each latent space of type $t$, the gating function $G_t(\cdot)$ determines the weight combinations used to aggregate the predictions from domain-specific classifiers $\{C_{t,k}(\cdot)\}_{k=1}^{K_t}$. The final prediction is obtained by averaging the predictions from all latent domains. The pseudocode of SLDG can be found in Appx. A.

## 4 Experiments

### 4.1 Experimental Setup

**Datasets.** We evaluate SLDG on two publicly available real-world EHR datasets: eICU [38] and MIMIC-IV [18], which are described as follows:

- **eICU** [38] covers over 200K visits for 139K patients admitted to the intensive care unit (ICU) in one of the 208 hospitals across the United States. The data was collected between 2014 and 2015. The 208 hospitals can be further categorized into four groups based on their location (Midwest, Northeast, West, and South). We use age, gender, and ethnicity as patient demographic information, and leverage the diagnosis, treatment, medication, and lab tables to gather patient visit information.
- **MIMIC-IV** [18] covers over 431K visits for 180K patients admitted to the ICU in the Beth Israel Deaconess Medical Center. The data was collected between 2008 to 2019. The approximate actual year of each admission is revealed as one of the four-year groups (2008-2010, 2011-2013, 2014-2016, and 2017-2019). We use age, gender, and ethnicity as patient demographic information, and leverage diagnoses, procedures, and prescriptions to gather patient admission information.

We elaborate on the cohort selection process and provide comprehensive dataset statistics in Appx. B.1. In the end, we extract 149227 visits from 116075 patients in the eICU dataset, and 353238 visits from 156549 patients in the MIMIC-IV dataset.

**Clinical Predictive Tasks.** We focus on two common clinical predictive tasks: (1) Readmission prediction, which aims to determine whether a patient will be readmitted within the next 15 days following discharge. (2) Mortality prediction, which aims to predict whether a patient will pass away upon discharge in the eICU setting, or within 90 days after discharge in the MIMIC-IV setting. A detailed explanation of this setting can be found in Appx. B.2.

**Data Split.** We evaluate the performance of our model across spatial gaps using the eICU dataset. For this purpose, we select the target testing data as the group (Midwest) that demonstrated the largest performance gap in a pilot study. The remaining groups (Northeast, West, and South) are used as the source training data. To assess the model's performance across temporal gaps, we utilize the MIMIC-IV dataset. Patients admitted after 2014 are used as the target testing data, while all preceding patients are included in the source training data. We elaborate more on the data split in Appx. B.3.

**Baselines.** We compare SLDG against three categories of baselines. (1) The first category consists of naive baselines, including **Oracle**, trained directly on the target data, and **Base**, trained solely on the source data. (2) The second category comprises DG methods that require domain IDs. These include **DANN [16]** and **MLDG [22]**, which use coarse regional and temporal groups as the domain definition, and **ManyDG [56]**, which treats each patient as a unique domain. (3) The last category consists of DG methods that do not rely on domain IDs, including **IRM [4]**, **MMLD [29]**, and **DRA [14]**. A detailed explanation of all the baselines can be found in Appx. B.4.

**Evaluation Metrics.** Both readmission prediction and mortality prediction are binary classification tasks. To evaluate the performance of the models, we calculate the Area Under the Precision-Recall Curve (AUPRC) and the Area Under the Receiver Operating Characteristic Curve (AUROC) scores. For each metric, we report the average scores and standard deviation by performing bootstrapping

Table 1: Results of domain generalization on the eICU and MIMIC-IV datasets. An asterisk (*) indicates that SLDG achieves a significant improvement over the best baseline method, with a p-value smaller than 0.05. The experimental results demonstrate that SLDG exhibits robustness against spatial (eICU) and temporal (MIMIC-IV) domain shifts.

| Method | eICU | | | | MIMIC-IV | | | |
|---|---|---|---|---|---|---|---|---|
| | Readmission | | Mortality | | Readmission | | Mortality | |
| | AUPRC | AUROC | AUPRC | AUROC | AUPRC | AUROC | AUPRC | AUROC |
| Oracle | 0.219 (0.01) | 0.677 (0.01) | 0.271 (0.01) | 0.839 (0.01) | 0.282 (0.01) | 0.693 (0.00) | 0.428 (0.00) | 0.898 (0.01) |
| Base | 0.104 (0.02) | 0.510 (0.01) | 0.230 (0.01) | 0.803 (0.01) | 0.237 (0.01) | 0.665 (0.01) | 0.374 (0.01) | 0.861 (0.00) |
| DANN | 0.135 (0.01) | 0.538 (0.01) | 0.245 (0.01) | 0.808 (0.01) | 0.247 (0.01) | 0.673 (0.01) | 0.380 (0.02) | 0.873 (0.02) |
| MLDG | 0.104 (0.01) | 0.525 (0.01) | 0.224 (0.01) | 0.797 (0.01) | 0.205 (0.01) | 0.637 (0.02) | 0.360 (0.01) | 0.857 (0.01) |
| ManyDG | 0.150 (0.01) | 0.549 (0.01) | 0.259 (0.01) | 0.814 (0.01) | 0.249 (0.01) | 0.676 (0.01) | 0.388 (0.01) | 0.880 (0.01) |
| IRM | 0.136 (0.01) | 0.538 (0.01) | 0.252 (0.02) | 0.811 (0.01) | 0.242 (0.00) | 0.668 (0.01) | 0.387 (0.01) | 0.876 (0.01) |
| MMLD | 0.167 (0.01) | 0.578 (0.00) | 0.256 (0.01) | 0.818 (0.01) | 0.250 (0.02) | 0.679 (0.01) | 0.393 (0.01) | 0.887 (0.01) |
| DRA | 0.148 (0.01) | 0.551 (0.01) | 0.249 (0.01) | 0.810 (0.01) | 0.246 (0.01) | 0.670 (0.01) | 0.387 (0.01) | 0.875 (0.01) |
| SLDG | **0.186 (0.01)\*** | **0.623 (0.01)\*** | **0.268 (0.01)\*** | **0.824 (0.01)\*** | **0.274 (0.01)\*** | **0.690 (0.01)\*** | **0.416 (0.00)\*** | **0.899 (0.01)\*** |

1000 times. Additionally, we conduct independent two-sample t-tests to assess whether SLDG achieves a significant improvement over the baseline methods.

**Implementation Details.** For all baselines, we use the same Transformer [48] architecture as the backbone encoder. Patient demographics features (age, gender, and ethnicity) are embedded with an embedding look-up table. We also embed the timestamps with sinusoidal positional encoding. The medical, patient demographics, and temporal embeddings are added together to form the overall sequence embedding. All models are trained for 100 epochs, and the best model is selected based on the AUPRC score monitored on the source validation set. For SLDG, UMAP [30] from UMAP-learn [41] is used with 2 components, 10 neighbors, and 0 minimum distance; and k-Means from Scikit-learn [35] is used with the default hyper-parameter. Further information regarding the detailed implementations can be found in Appx. B.5.

## 4.2 Main Results

Table 1 presents the domain generalization results on the eICU [38] and MIMIC-IV [18] datasets. Firstly, we observe a significant performance gap between the Oracle and Base methods, indicating the presence of substantial spatial and temporal domain gaps. This supports the use of the DG setting. Notably, the readmission tasks exhibit larger domain gaps, which is reasonable since hospitals across different locations and timestamps may have varying criteria for patient readmission. Secondly, we note that the two DG methods, DANN [16] and MLDG [22], utilizing coarse domain partitions such as region and timestamp, achieve minimal or no improvements. This outcome is expected because the domain partitions are too coarse, making it challenging to identify consistent domain features. In comparison, ManyDG [56] achieves better performance by considering each individual patient as a unique domain. Among the remaining three baseline methods that do not rely on domain IDs, IRM [4] demonstrates the slightest improvement. DRA [14] performs better due to the usage of multi-head networks, which share a similar intuition as SLDG. MMLD [29] attains the highest performance among all baselines, showcasing the advantages of explicit domain discovery. Lastly, SLDG outperforms baselines for all tasks. Specifically, in terms of the AUPRC score, SLDG achieves an 11% relative improvement in eICU readmission prediction, 3% in eICU mortality prediction, 10% in MIMIC-IV readmission prediction, and 6% in MIMIC-IV mortality prediction.

## 4.3 Quantitative Analysis

This section provides quantitative analyses to elucidate the performance enhancements achieved by SLDG. The analyses encompass the evaluation of clustering results, ablation studies on the clustering algorithm, the impact of the number of clusters and iterations, and a runtime comparison.

**Evaluation of clustering results.** First, we evaluate the domain recovery ability of DG methods that do not rely on domain IDs, namely MMLD [29], DRA [14], and the proposed SLDG. Since the actual latent domain categorizations are unavailable, we assess the separability of the learned clustering results with the Silhouette score [40]. Note that the reported Silhouette score is calculated on the testing set, while the hyper-parameters are chosen based on the Silhouette score on the training set. As depicted in Fig. 3, DRA achieves the lowest score, which aligns with expectations as it solely learns latent domain categorizations through multi-head networks without explicit clustering. In contrast, MMLD generates more distinct clusters due to its iterative clustering and training setup.

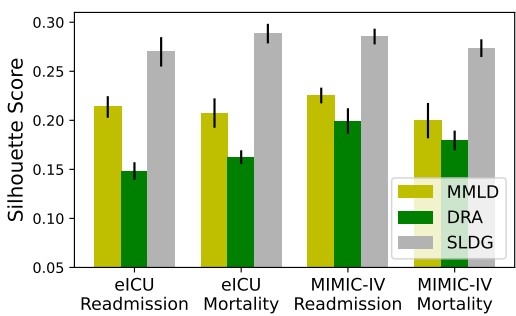

Figure 3: Performance of latent domain clustering. A higher Silhouette score indicates improved cluster separability.

However, it still necessitates manual specification of the number of clusters. In comparison, SLDG obtains the highest score using an automated hierarchical clustering technique.

**Influence of the clustering algorithm.** Next, we assess the influence of different clustering algorithms. Naive k-Means and GMM require manual specification of the number of clusters, which we set to the same value as SLDG's. The results can be found in Tab. 2. We observe that naive k-Means and GMM achieve similar perform similarly to the best baseline methods in Tab. 1. This outcome is reasonable since the success of SLDG relies on both the accurate discovery of latent domains and customized models for each domain. Naive clustering techniques often fail to identify subtle yet important latent domains. In contrast, SLDG, utilizing the automatic hierarchical clustering technique, achieves the highest score.

Table 2: Ablation study on the influence of the clustering algorithm.

| Method | eICU | | | | MIMIC-IV | | | |
| | Readmission | | Mortality | | Readmission | | Mortality | |
| | AUPRC | AUROC | AUPRC | AUROC | AUPRC | AUROC | AUPRC | AUROC |
|---|---|---|---|---|---|---|---|---|
| SLDG + k-Means | 0.148 (0.01) | 0.553 (0.01) | 0.249 (0.01) | 0.814 (0.00) | 0.250 (0.01) | 0.670 (0.01) | 0.388 (0.01) | 0.886 (0.01) |
| SLDG + GMM | 0.143 (0.01) | 0.549 (0.01) | 0.240 (0.01) | 0.808 (0.01) | 0.250 (0.01) | 0.688 (0.01) | 0.390 (0.00) | 0.888 (0.00) |
| SLDG | **0.186 (0.01)*** | **0.623 (0.01)*** | **0.268 (0.01)*** | **0.824 (0.01)*** | **0.274 (0.01)*** | **0.690 (0.01)*** | **0.416 (0.01)*** | **0.899 (0.01)*** |

**Influence of the number of clusters and iterations.** Next, we analyze the impact of the number of clusters and iterations on the eICU readmission prediction task. We also compare our results with the best-performing baseline, MMLD [29]. The results can be found in Fig. 4. The upper panel of the figure shows that the model's performance initially improves with an increasing number of clusters. This improvement can be attributed to the finer granularity of clustering, which enables better identification of domains and customization of experts. However, as the number of clusters continues to increase, the model's performance starts to decline. This decline is caused by the growing number of model parameters, making training more challenging and leading to overfitting on suspicious samples. Similarly, the trend observed in the lower panel of the figure for the number of iterations aligns with the number of clusters. The performance initially improves as

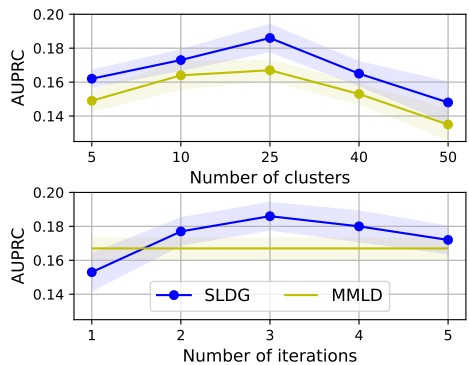

Figure 4: Results on the influence of the number of clusters and iterations.

the number of iterations increases, allowing the model to learn more from the data. However, after a certain point, the model starts overfitting on specific clusters, leading to decreased performance.

**Runtime comparison.** Lastly, we compare the training time of SLDG with the naive Base baseline. All runtimes are measured on a single NVIDIA A6000 GPU. The results can be found in Tab. 3. The use of the UMAP [30] dimensionality reduction technique enables SLDG to perform clustering quickly. As a result, the training time overhead of SLDG is reasonably low overall (18% on eICU and 20% on MIMIC-IV) compared to the significant performance improvement achieved (up to 79% relative improvement on AUPRC score on eICU and up to 15% on MIMIC-IV).

Table 3: Runtime comparison.

| Method | eICU | MIMIC-IV |
|---|---|---|
| Base | 67 min | 93 min |
| SLDG | 79 min | 112 min |

## 4.4 Case Study

| Latent Space | Frequent Clinical Events from the Top-2 Clusters |
|---|---|
| Diagnosis | Sepsis, Infection, Kidney failure |
| | Cardiac Arrest, Congestive Heart Failure |
| Treatment | Heart valve procedures, Cardiovascular monitoring |
| | Ventilation, Respiratory intubation |
| Medication | Propofol, Lorazepam, Fentanyl |
| | Amiodarone, Noradrenaline |

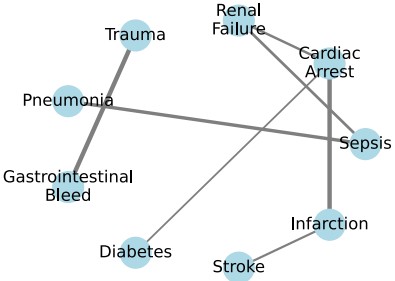

Figure 5: Left: Common clinical events observed in two largest domains identified from each latent space. Right: Learned similarity among the identified domains within the diagnostic latent space.

For the case study, we first examine the recovered domains within each latent space. The results are presented in the left panel of Fig. 5. It is evident that the common events identified within the same domains are consistent. For instance, *propofol*, *lorazepam*, and *fentanyl* is frequently used together, serving the purpose of anesthesia (pre-surgery), sedation (in-surgery), and pain management (post-surgery). Furthermore, the right panel of Figure 5 illustrates a visualization of the learned domain similarity, i.e., the distances between the domain prototypical weights $\{\mathbf{w}_{t,k}\}_{k=1}^{K_t}$. Notably, our method (SLDG) successfully captures meaningful relationships among the latent domains. For instance, a strong relationship is learned between *pneumonia* and *sepsis*. In clinical practice, when pneumonia is severe or if the infection spreads beyond the lungs, it can enter the bloodstream and trigger a systemic response, leading to sepsis [8]. Another example is the observed strong association between *cardiac arrest* and *infarction*. In practice, if a significant portion of the heart muscle is damaged, it can disrupt the heart's electrical system, potentially leading to cardiac arrest [19].

## 5 Related Work

**Domain Generalization** The goal of DG is to learn a model using data from multiple source domains in order to achieve effective generalization to a distinct target domain [7]. To achieve this, domain alignment approaches try to match the feature distributions among multiple source domains with techniques such as moments minimization [32, 25], contrastive learning [31], adversarial learning [25, 45], regularizers [4, 24], and augmentation [62, 50, 44, 60]. Meta-learning frameworks have also been utilized to simulate new domain scenarios during training [23, 27, 26, 5, 22, 29]. Additionally, domain-specific model ensemble techniques have been employed [9, 42, 43, 61]. However, these conventional DG methods assume the availability of domain IDs, which may not be feasible in healthcare settings where patients can belong to numerous unobserved domains.

Recent advancements in DG have attempted to alleviate the reliance on domain IDs [29, 62, 14, 10, 33]. MMLD [29] is the most relevant prior work to ours. It simultaneously discovers latent domains and learns domain-invariant features through adversarial learning. However, it focuses on training a

single model that generalizes across all domains. Given that patients from different domains exhibit distinct characteristics and require different treatment approaches [2, 58], training a single model for all domains poses challenges and can result in sub-optimal performance.

**Clinical Predictive Modeling.** The main objective of clinical predictive modeling is to predict the occurrence of future events, such as 15-day hospital readmission and 90-day mortality, based on existing patient information. Deep learning models have been widely used in clinical predictive modeling with EHR data [55, 57]. These models are designed to capture temporal patterns in patient data [11, 36, 6, 28], model structural information in medical codes [13, 52], augment the model using pre-training [39], or leverage patient similarities for better decision making [58, 2]. However, these models typically assume an unchanged test domain and may suffer from degraded performance with domain shift. To address this issue, AutoMap [53] solves the feature space shift issue by learning an auto-mapping function without considering any distribution shift. MedLink [54] aggregates de-identified patient data from different sites to enable joint training. ManyDG [56] tackles patient covariate shift by treating each patient as a unique domain and disentangling domain variant and invariant features. However, maintaining a large number of domains is unnecessary, as similar patients often exhibit similar clinical behavior and can share a common domain.

# 6 Conclusion

Clinical predictive models often exhibit degraded performance when applied to data from new regions or future periods due to distribution shifts. To address this, we propose SLDG, a self-learning framework that iteratively identifies decoupled domains and trains customized classifiers for each domain. We evaluate SLDG on two medical datasets, and our results show that it outperforms all baseline methods. In addition, we provide detailed qualitative analyses and case studies to support our findings.

# Acknowledgments

This work was supported by NSF award SCH-2205289, SCH-2014438, and IIS-2034479. This project has been funded by the Jump ARCHES endowment through the Health Care Engineering Systems Center.

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

## Contents of Appendix

# A    Pseudocode of `SLDG`

---

**Algorithm 1:** Training and Inference for `SLDG`.

---

1: *// Training*

**Require:** Source training data from $P_{tr}$

2: Pre-train patient encoder $E(\cdot)$ on the same task with binary cross-entropy loss for 40 epochs

3: **for** iteration ranging from 1 to 3 **do**

4:     Perform decoupled domain discovery with the encoder $E(\cdot)$ by Eq. (2), (3)

5:     Initialize gating and classifier weights $\mathbf{w}_{t,k}, \mathbf{w}_{t,k}^+, \mathbf{w}_{t,k}^-$ by Eq. (10), (8)

6:     **for** epoch ranging from 1 to 20 **do**

7:         **for** each patient $(x, y) \sim P_{tr}$ **do**

8:             Obtain decoupled patient representations $\{\mathbf{h}_t\}_{t \in \mathcal{T}}$ by Eq. (4), (5)

9:             Compute domain-specific predictions $C_{t,k}(\mathbf{h}_t)$ by Eq. (7)

10:             Compute gating weights $G_t(\mathbf{h}_t)$ by Eq. (9)

11:             Obtain final prediction $o$ by Eq. (6)

12:             Update model parameters with binary cross-entropy loss

13:         **end for**

14:     **end for**

15: **end for**

16: *// Inference*

**Require:** Target testing data from $P_{te}$

17: **for** each patient $(x, y) \sim P_{tE}$ **do**

18:     Obtain decoupled patient representations $\{\mathbf{h}_t\}_{t \in \mathcal{T}}$ by Eq. (4), (5)

19:     Compute domain-specific predictions $C_{t,k}(\mathbf{h}_t)$ by Eq. (7)

20:     Compute gating weights $G_t(\mathbf{h}_t)$ by Eq. (9)

21:     Obtain final prediction $o$ by Eq. (6)

22: **end for**

---

# B    Additional Experimental Setup

## B.1    Datasets

For both datasets, we select our cohorts by filtering out visits of patients younger than 18 or older than 89 years old, visits that last longer than 10 days, and visits with data from less than 3 or more than 256 timestamps. In the case of the eICU dataset, we additionally exclude visits lasting shorter than 12 hours, as the predictions are made 12 hours after admission. Similarly, for the MIMIC-IV dataset, we exclude visits where the patient ultimately passed away, as the predictions are made upon discharge. Tab. 4 provides detailed statistics of the two datasets.

## B.2    Clinical Predictive Tasks

We focus on two common clinical predictive tasks: readmission prediction and mortality prediction.

In the case of the eICU dataset, the predictions are made 12 hours after admission. Readmission prediction aims to determine whether a patient will be readmitted within the next 15 days following discharge. Mortality prediction, on the other hand, aims to predict whether a patient will pass away upon discharge. The overall prevalence for these tasks is 15% for readmission and 4% for mortality.

For the MIMIC-IV dataset, the predictions are made at the time of discharge. Similar to the eICU dataset, the readmission prediction task is defined as predicting whether a patient will be readmitted within 15 days after discharge. To prevent information leakage, the mortality prediction task for MIMIC-IV is defined as predicting whether a patient will pass away within 90 days after discharge. The overall prevalence for these tasks is 14% for readmission and 4% for mortality.

Table 4: Dataset statistics.

| Item | eICU | MIMIC-IV |
|---|---|---|
| #Patients | 116075 | 156549 |
| #Admissions | 149227 | 353238 |
| Readmission Rate | 0.15 | 0.14 |
| Mortality Rate | 0.04 | 0.04 |
| **Region: Midwest** | | **Year: 2008-2010** |
| #Patients | 29767 | 37328 |
| #Admissions | 35989 | 56433 |
| Readmission Rate | 0.10 | 0.14 |
| Mortality Rate | 0.03 | 0.04 |
| Age | 62 | 56 |
| Gender | F: 0.46, M: 0.54 | F: 0.53, M: 0.47 |
| Race | African American: 0.09, Asian: 0.01, Caucasian: 0.83, Hispanic: 0.01, Native American: 0.01, Other: 0.04 | African American: 0.15, Asian: 0.03, Caucasian: 0.71, Hispanic: 0.06, Native American: 0.00, Other: 0.04 |
| Average #Events | 90.01 | 31.87 |
| **Region: Northeast** | | **Year: 2011-2013** |
| #Patients | 5886 | 39125 |
| #Admissions | 6958 | 62586 |
| Readmission Rate | 0.17 | 0.15 |
| Mortality Rate | 0.06 | 0.04 |
| Age | 62 | 57 |
| Gender | F: 0.44, M: 0.56 | F: 0.53, M: 0.47 |
| Race | African American: 0.03, Asian: 0.01, Caucasian: 0.92, Hispanic: 0.01, Native American: 0.00, Other: 0.03 | African American: 0.17, Asian: 0.03, Caucasian: 0.66, Hispanic: 0.07, Native American: 0.00, Other: 0.07 |
| Average #Events | 104.54 | 35.19 |
| **Region: South** | | **Year: 2014-2016** |
| #Patients | 27584 | 41737 |
| #Admissions | 33033 | 64592 |
| Readmission Rate | 0.11 | 0.14 |
| Mortality Rate | 0.04 | 0.04 |
| Age | 62 | 57 |
| Gender | F: 0.46, M: 0.54 | F: 0.52, M: 0.48 |
| Race | African American: 0.21, Asian: 0.01, Caucasian: 0.68, Hispanic: 0.05, Native American: 0.00, Other: 0.03 | African American: 0.17, Asian: 0.04, Caucasian: 0.66, Hispanic: 0.06, Native American: 0.00, Other: 0.07 |
| Average #Events | 84.28 | 36.53 |
| **Region: West** | | **Year: 2017-2019** |
| #Patients | 17670 | 40496 |
| #Admissions | 19803 | 63654 |
| Readmission Rate | 0.29 | 0.14 |
| Mortality Rate | 0.04 | 0.04 |
| Age | 63 | 58 |
| Gender | F: 0.45, M: 0.55 | F: 0.52, M: 0.48 |
| Race | African American: 0.05, Asian: 0.03, Caucasian: 0.77, Hispanic: 0.05, Native American: 0.02, Other: 0.08 | African American: 0.17, Asian: 0.04, Caucasian: 0.65, Hispanic: 0.06, Native American: 0.00, Other: 0.07 |
| Average #Events | 85.29 | 36.95 |

## B.3 Data Split

The eICU dataset comprises data collected from hospitals across the United States, while the MIMIC-IV dataset spans a period of ten years. Therefore, we utilize the eICU dataset to evaluate the model's performance across spatial gaps, and the MIMIC-IV dataset to assess its performance across temporal gaps.

For the eICU dataset, we divide it into four spatial groups based on regions: Midwest, Northeast, West, and South. Each group is then split into 70% for training, 10% for validation, and 20% for testing. We evaluate the gap between groups by comparing the performance of the backbone model trained on data from within the same group and data from outside the group. The target testing data is selected as the group (Midwest) that exhibits the largest performance gap, while the remaining groups (Northeast, West, and South) are used as the source training data.

Regarding the MIMIC-IV dataset, we divide it into four temporal groups: 2008-2010, 2011-2013, 2014-2016, and 2017-2019. Each group is further split into training, validation, and testing sets with a ratio of 70%, 10%, and 20% respectively. We consider patients admitted after 2014 as the target testing data, while all preceding patients are included in the source training data.

### B.4  Baselines

We first compare `SLDG` to two naive baselines.

- **Oracle:** We directly train a backbone model on the training set of the target domain, select the best model on the target validation set, and evaluate its performance on the target testing set. This model is trained with in-domain data and can be seen as a upper bound for all domain generalization method.

- **Base:** We train a backbone model on the training set of the source domain, select the best model on the source validation set, and evaluate its performance on the target testing set. This model is trained with out-domain data and should act as a performance lower bound.

We then compare `SLDG` to both classic and recent domain generalization methods. For a fair comparisons, all the methods below are trained on the source training set, selected on the source validation set, and tested on the target testing set.

- **DANN [16]:** Domain-Adversarial Neural Networks leverage a domain classifier and a gradient reversal layer to extract domain-invariant representations. This method uses the coarse regional and temporal groups as the domain definition.

- **MLDG [22]:** Meta-Learning for Domain Generalization adopts the Model-Agnostic Meta-Learning (MAML) [15] framework and simulates the new domain scenario during training. This method also uses the coarse regional and temporal groups as the domain definition.

- **ManyDG [56]:** Many-Domain Generalization disentangles domain-variant and invariant features through mutual reconstruction and orthogonal projection. This method treats each patient as a unique domain.

- **IRM [4]:** Invariant Risk Minimization learns domain-invariant representations by minimizing a bound on the expected generalization error under domain shifts. It acts as a regularizer and does not require domain IDs.

- **MMLD [29]:** Domain Generalization using a Mixture of Multiple Latent Domains iteratively assigns pseudo domain labels via clustering and trains the domain-invariant feature extractor through adversarial learning. This method does not rely on domain IDs.

- **DRA [14]:** Latent Domain Learning with Dynamic Residual Adapters uses layer-wise multi-head correction networks with a gating mechanism and residual connection to enhance model learning. This method does not rely on domain IDs.

### B.5  Implementation Details

For all baselines, we use the Transformer as the backbone encoder. The number of layers is 3, the embedding dimension is 128, the number of attention heads is 2. The event embedding look-up table is initialized with ClinicalBERT [3] embeddings of the event name and then project it down to 128 dimension with a linear layer. Patient demographics features (age, gender, and ethnicity) are separately embeded with another embedding look-up table. We also embed the timestamps with sinusoidal positional encoding. The medical, patient demographics, and temporal embeddings are added together to form the overall sequence embedding. For `SLDG`, UMAP [30] from UMAP-learn [41] is used with 2 components, 10 neighbors, and 0 minimum distance; and k-Means from Scikit-learn [35] is used with the default hyper-parameter. We apply a dropout of rate 0.2. We use Adam as the optimizer with a learning rate of 1e-4 and a weight decay of 1e-5. All models are trained for 100 epochs. The batch size is 256. We select the best model by monitoring the AUPRC score on the source validation set (except for the Oracle baseline, where we directly use the target validation set). We implement `SLDG` using PyTorch [34] 1.11 and Python 3.8. The model is trained on a CentOS Linux 7 machine with 128 AMD EPYC 7513 32-Core Processors, 512 GB memory, and eight NVIDIA RTX A6000 GPUs.

## C  Limitations and Broader Impacts

In terms of limitations, it is important to acknowledge that our work operates under the assumption that the target testing data still exhibit some similarities with the source training data. If there is

a significant distribution shift, the knowledge acquired from the source training data may become irrelevant. In such cases, neither the DG baselines nor our proposed method can effectively address the problem. It would be more appropriate to explore transfer learning or train a new model to obtain a better solution. Further, we propose SLDG to tackle two main challenges: (1) unknown domain IDs and (2) distinct characteristics across domains. In the scenario when the domain IDs are given and clearly separable (e.g., photo, art painting, cartoon, and sketch in the PACS [21] dataset), SLDG 's domain discovery approach might be unnecessary. Existing DG methods directly utilizing the domain IDs could be a better solution.

In terms of broader impacts, our work tackles a practical and prevalent issue in healthcare known as the domain shift problem. We aim to inspire future research in this area: both by investigating the existence of domain shift under various scenarios, and by contributing to the development of effective solutions for this real-world challenge.

## D Notations

| Notation | Meaning |
|---|---|
| $x$ | a patient's hospital visit |
| $[e_1, \ldots, e_m]$ | sequence of $m$ events |
| $t$ | type of an event |
| $T(\cdot)$ | mapping function from event to its type |
| $\mathcal{E}$ | set of all events |
| $\mathcal{T}$ | set of all event types |
| $y \in \{+, -\}$ | label, i.e., the occurrence of a certain future event |
| $f_\phi(\cdot)$ | overall clinical predictive model |
| $\phi$ | model parameter |
| $l(\cdot)$ | loss function |
| $P_{tr}, P_{te}$ | training and testing data distribution |
| $E_t(\cdot)$ | feature-specific patient encoder for event type $t$ |
| $\mathbf{h}_t$ | patient representation in latent space of type $t$ |
| $h$ | hidden dimension |
| $\{\mathbf{h}_t^{(i)}\}_{i=1}^{N_{tr}}$ | all patient representations in latent space of type $t$ |
| $N_{tr}$ | total number of training samples |
| $K_t$ | number of discovered domains in the latent space of type $t$ |
| $\mathbf{M}_t$ | domain assignment matrix |
| $[\mathbf{e}_1, \ldots, \mathbf{e}_m]$ | contextualized representation for event sequence $[e_1, \ldots, e_m]$ |
| $E(\cdot)$ | embedding function |
| $\{\mathbf{h}_t\}_{t \in \mathcal{T}}$ | multi-vector representations for a single patient |
| $C_{t,k}(\cdot)$ | customized classifier for the discovered domain $k$ in the latent space of type $t$ |
| $G_{t,k}(\cdot)$ | the gating weight for the customized classifier $C_{t,k}(\cdot)$ |
| $o$ | model output |
| $\mathbf{w}_{t,k}^+, \mathbf{w}_{t,k}^-$ | learnable prototype weight vectors of the positive and negative classes for the $k$-th discovered domain in the latent space of type $t$ |
| $d(\cdot, \cdot)$ | Euclidean distance |
| $\mathbf{w}_{t,k}$ | learnable prototypical weight vector for the discovered domain $k$ in the latent space of type $t$ |

## E Additional Illustrations

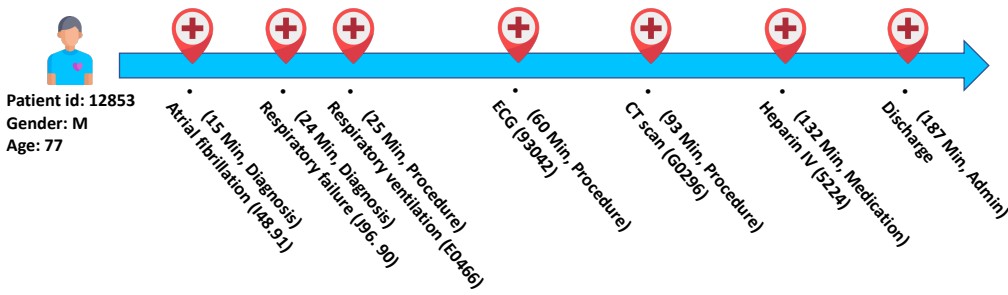

Figure 6: An illustration of the patient visit as input.

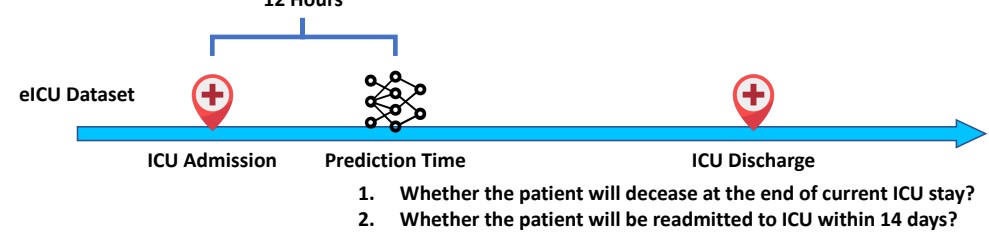

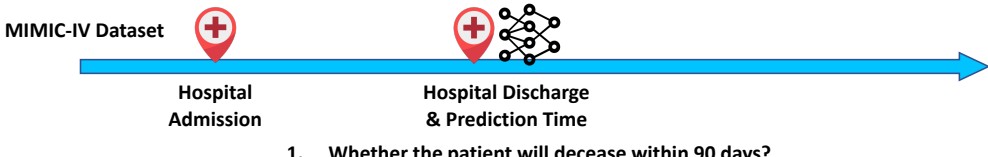

Figure 7: An illustration of the task definitions in the eICU and the MIMIC-IV datasets.

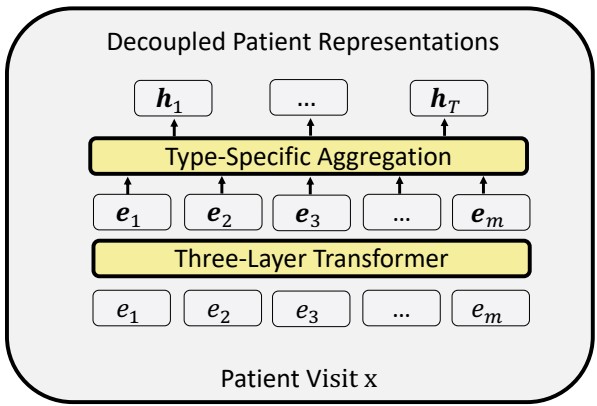

Figure 8: The architecture of the feature-specific patient encoder.

