# OpenReview forum: "An Iterative Self-Learning Framework for Medical Domain Generalization"
_NeurIPS.cc/2023/Conference — NeurIPS 2023 poster_

### Official Review · Reviewer_1qdB · 2023-07-05

**Soundness:** 2 fair
**Presentation:** 4 excellent
**Contribution:** 4 excellent
**Rating:** 7
**Confidence:** 3

**Summary:**

In this work, the authors present SLDG: a framework for dealing with domain shifts in clinical event data. They propose two steps in this framework. First, an unsupervised latent-domain clustering which automatically clusters different subjects into different feature-level clusters via monitoring the Silhouette score for optimal clusters, followed by individually trained classifiers for each cluster group, where the final probabilistic prediction is the weighted combination of the predictions from within each cluster group. The individually trained classifier is the softmax-ed distance between a patient's latent representation and the prototypical representations for the latent space. The authors find that by training this method by reinitializing the clusters every set number of epochs they can beat existing domain adaptation methods.

**Strengths:**

- The idea and method are communicated in a clear manner and provide a good understanding of the underlying problem.
- The method has clear benefits over baseline methods, sometimes with large jumps in improvement (Readmission on eICU).
- The work does offer a unique and interesting method for self-supervised clustering of latent features. I really liked the proposed method and thought that it was well explained.
- The ablation of the clustering algorithm demonstrates that the added complexity of the hierarchical clustering indeed shows benefits over traditional methods, something which other works could potentially build on top of.


**Weaknesses:**

- Customized classifiers for each domain are a very inefficient method of scaling to new domains even with the efficient parameterization of each domain. Although, this is definitely more problematic if the latent dimension is larger or smaller and there doesn't seem to be any kind of ablation for the size of the classifier dimension. I believe this might be a useless result to have.
- The formulation of the prototypical vectors I don't think is very well motivated and is confusing why there is a positive and negative representation of each latent space t. Additionally, I'm not exactly sure why it would make sense to average the predictions from all latent domains to make new predictions given that for the downstream prediction task, not all clusters are probably equally helpful with making predictions. This seems like it would introduce a lot of noise into predictions.
- The biggest weakness I find with the analysis of this work is that it is unclear how large the baseline methods are in comparison to the proposed method. The paper lacks an analysis of parameter numbers which is problematic because as you develop more clusters, it is my understanding that you will also balloon up your parameter count, but also we don't know how big the transformer architecture is in comparison to other baseline methods.


**Questions:**

- What is meant on line 121 with the embedding lookup table?
- What is the intuition for using sinusoidal positional embeddings? I mean to say, why does it make sense here to embed timestamps with sinusoidal positional encodings when the patient visits would only be going forwards in time? Maybe I misunderstood this piece and would need some clarification.
- The method for reducing the clusters from exponential to linear makes sense, but shouldn't the label types be coupled to some degree? Many of those features shouldn't be independent of each other so making an independence assumption is strange.
- Related to my point of weakness with evaluation, I would definitely consider raising my score if there was additional necessary context concerning the capacity of the proposed method (with additional parameters per clustering included) in comparison to the baseline methods. I think that the clustering method is sufficiently novel and interesting that it would be helpful for others to know about, the only thing keeping my score at a borderline except at the moment is that I am unsure if the comparison to the baselines is fair.


**Limitations:**

The authors don't explicitly discuss the potential negative social impact of their work in the main body of the text, right now it is in the appendix, and I believe that it should be moved to the main body. It does address, however, the limitations of the work and places it meaningfully in context.

---

> ### Author Rebuttal · Authors · 2023-08-10
>
>
> We would like to thank the reviewer for the positive feedback. We have made the necessary updates to our manuscript. Below we would like to take this opportunity to respond to your questions.
>
>
> **Q1. What is meant on line 121 with the embedding look-up table?**
>
> A1. The embedding look-up table is essentially a mapping from the event to a static embedding vector. For example, for the event “acute heart failure”, we obtain its static embedding by feeding the string “acute heart failure” through the ClinicalBERT and using the pooled sequence representation as the event embedding (dimension 768). In practice, we stack the embeddings for all events together into a matrix and store it in the `nn.Embedding()` layer.
>
>
> **Q2. What is the intuition for using sinusoidal positional embeddings?**
>
> A2. In both datasets, each event is associated with a timestamp denoting the time when the event occurred. We transform the absolute time to the relative time after patient admission (in hours) and embed the hour value using sinusoidal positional encoding. The intuition behind the temporal embedding is that the timestamps can provide an important indication of the speed of the patient's health progression (e.g., acute v.s. chronic).
>
>
> **Q3. The label types can be coupled to some degree?**
>
> A3. We wish to clarify the confusion regarding our feature-specific patient encoding module. Specifically, for a patient’s hospital visit $x = [e_1, . . . , e_m]$, we first pass it through a Transformer encoder to obtain the contextualized event representations, as $[ \mathbf{e}_1, \dots, \mathbf{e}_m ] = E ( [ e_1, \dots, e_m ] )$. The Transformer encoder employs a self-attention mechanism to capture relationships among event pairs. Subsequently, we aggregate the contextualized event representations based on their respective types. This aggregation results in a distinct patient representation that takes into account different medical codes. Consequently, each medical code contributes to the interaction with all other codes within the same patient visit.
>
> **Q4. The capacity of the proposed method (with additional parameters per clustering included) in comparison to the baseline methods.**
>
> A4. The main computation overhead of SLDG happens during the training phase, where we need to perform an additional hierarchical clustering step. However, the computation overhead is minimized due to the usage of UMAP dimension reduction. As shown at the end of section 4.3, the training time of SLDG only increases by 12 minutes compared to the most efficient base method. In terms of inference efficiency and model complexity, SLDG is very similar to the other DG methods not requiring domain IDs (i.e., MMLD and DRA). Below we compare model parameters and inference efficiency between these methods.
>
> *Table: Additional comparison of model parameter size and inference time.*
>
> | Method | \# Params | Inference Time (ms) |
> |---|---|---|
> | Base | 1.3M | 33 |
> | MMLD | 1.5M | 54 |
> | DRA | 1.5M | 63 |
> | SLDG (Ours) | 1.5M | 68 |
>
> Thank you for taking the time during the rebuttal phase. We hope that these updates address your questions.

---

> > ### Comment · Reviewer_1qdB · 2023-08-11
> > **Thank you for the clear response!**
> >
> > I thank the authors for their clear response, in reflection I will be raising my score to an accept because I am satisfied with the analysis of parameter counts and inference time measurements. I think that both this analysis and the clarification regarding the feature-specific patient encoding module would be a necessary step for this paper being better placed in context and should be added.

---

> > > ### Author Response · Authors · 2023-08-17
> > >
> > > Thank you for considering our response and deciding to raise the score! We appreciate the valuable feedback and will include the analysis of parameter counts and inference time, and clarifications regarding the feature-specific patient encoding module in the final version. Please do not hesitate to let us know if you have any other questions. Thanks!

---

### Official Review · Reviewer_36fo · 2023-07-09

**Soundness:** 3 good
**Presentation:** 2 fair
**Contribution:** 3 good
**Rating:** 6
**Confidence:** 3

**Summary:**

the authors present a novel self-learning framework, sldg, designed to address the domain shift problem in deep learning models, particularly in the context of clinical decision-making. the authors posit that reliance on domain ids is a key limitation in existing domain generalization algorithms. to overcome this, the sldg framework iteratively decouples domains and trains personalized classifiers for each domain. the results demonstrate improvements in the auprc score over baselines.

edit august 19, 2023: increase score from 5 to 6 in light of authors' rebuttal.

**Strengths:**

1. **innovative approach to domain generalization**: the sldg framework introduces a novel way to handle the domain shift problem in deep learning models. unlike existing methods that rely on domain ids, sldg iteratively discovers decoupled domains and trains personalized classifiers for each domain. the approach is clearly explained.

2. **significant performance improvement**: the paper demonstrates that the sldg framework can achieve a substantial improvement in the auprc score over the best baseline, indicating the promise of the proposed method.

3. **applicability in healthcare settings**: the authors have focused on the application of their model in clinical decision-making, a field where domain shift is a common issue. the ability of the sldg framework to handle potential domain shifts when applied to target data is a significant strength, as it makes the model highly relevant and potentially impactful in real-world healthcare settings.

4. **comprehensive technical details**: the inclusion of a detailed pseudocode for the sldg algorithm in the appendix is a strength. it provides a clear understanding of the steps involved in training and inference, which can be beneficial for researchers and practitioners looking to implement or build upon this work.

5. **robust testing**: the authors have tested their algorithm on two different datasets for readmission and mortality prediction tasks. this robust testing adds credibility to the results and demonstrates the versatility of the algorithm. the filtering of datasets to exclude certain visits also shows a thoughtful approach to ensuring the quality and relevance of the data used.

**Weaknesses:**

1. **domain discovery process**: the paper presents an iterative process for decoupled domain discovery (section 3.2). a detailed explanation of how the algorithm ensures the discovered domains are meaningful and distinct from each other. a more thorough discussion or a visual representation of the discovered domains could enhance the understanding of this process.

2. **baseline comparison**: the paper mentions that sldg outperforms the best baseline (section 4.1). however, many related approaches are not mentioned and no explanation is provided why they should be discarded or how they relate to sldg. including more comparisons, empirically or at least conceptually, with state-of-the-art models could strengthen the paper's claims about sldg's novelty. a good general overview can be found at pwc https://paperswithcode.com/task/domain-generalization.

3. **algorithm complexity**: the pseudocode provided in the appendix outlines the steps of the sldg algorithm, but the paper does not discuss the computational complexity of these steps beyond base and sldg. providing runtime information for the other dgs as well as flops or macs information could help readers understand the feasibility of implementing sldg in practice.

4. **generalizability**: while the paper demonstrates the effectiveness of sldg in healthcare settings, it does not discuss its applicability to other fields. a discussion on how the proposed method could be adapted for other domains would increase the paper's impact and reach. similarly, the title suggests application to medicine broadly, while experiments are performed on ehr data only.

5. **relation to connected related fields**: the authors do not discuss in detail and only briefly at the end how the proposed sldg framework relates to or differs from other relevant techniques. some important ones, such as federated learning and disentangled representation learning, are not mentioned at all despite similar problems having been tackled in these fields (see for example https://ieeexplore.ieee.org/abstract/document/9174890 or https://aclanthology.org/P19-1041/, among many other works*). given that these methods also deal with domain generalization, often without domain ids such as in private distributed clients, a comparison or discussion could provide valuable context and highlight the unique contributions of sldg.

*i have no affiliation to examples of related work

**Questions:**

1. can you elaborate on the process of decoupled domain discovery in sldg?

2. how could the meaningfulness of each domain sldg identifies be evaluated more thoroughly?

3. why were certain related approaches not included in the comparison with sldg?

4. could you provide a more comprehensive contextualization with existing dg methods?

5. could you provide more details on the computational complexity of sldg vis-a-vis the other dg baselines you employ in your experiments?

6. the experiments seem focused on ehr data, but the title suggests broader applicability. could you clarify?

7. how would sldg be adapted for domains beyond ehr?

8. how does sldg relate to techniques like federated learning with clients that have non-iid data?

9. how does sldg relate to disentangled representation learning?

10. these methods also tackle domain generalization, often without domain ids. could you discuss any potential connections or distinctions?

**Limitations:**

it would be great to discuss more connections to related work from connected disciplines working on similar problems and include a clear discussion of limitations in the main text, which is missing at the moment.

---

> ### Author Rebuttal · Authors · 2023-08-10
>
> We would like to thank the reviewer for the positive feedback. We have made the necessary updates to our manuscript. Below we would like to take this opportunity to respond to your questions.
>
>
> **Q1. Can you elaborate on the process of decoupled domain discovery and the meaningfulness of each domain?**
>
> A1. The hierarchical domain clustering is only applied to the training set during model training. Specifically, we first use the UMAP algorithm to reduce the embedding dimension of each sample to 2. Then, we run a hierarchical k-Means algorithm to cluster the samples in the dimension-reduced latent space. During testing, the test sample is mapped to different clusters based on the distance between its embedding and the mean cluster embeddings. In section 4.4, we provide a case study and visualization regarding the meaningfulness of the discovered domain.
>
>
> **Q2. Additional comparison with existing dg method.**
>
> A3. Below, we provide additional experiments with two recent DG methods: GroupDRO [2] and GMoE [3].
>
> *Table: Comparison between SLDG and two additional DG methods. The SLDG archives the best performance in all tasks.*
>
> | Method | eICU readmission | eICU mortality | MIMIC-IV readmission | MIMIC-IV readmission |
> |---|---|---|---|---|
> | GroupDRO | 0.158 | 0.252 | 0.244 | 0.397 |
> | GMoE | 0.169 | 0.246 | 0.248 | 0.385 |
> | SLDG (Ours) | **0.186** | **0.268** | **0.274** | **0.416** |
>
>
> **Q3. Computational complexity of SLDG.**
>
> A3. The main computation overhead of SLDG happens during the training phase, where we need to perform an additional hierarchical clustering step. However, the computation overhead is minimized due to the usage of UMAP dimension reduction. As shown at the end of section 4.3, the training time of SLDG only increases by 12 minutes compared to the most efficient base method. In terms of inference efficiency and model complexity, SLDG is very similar to the other DG methods not requiring domain IDs (i.e., MMLD and DRA). Below we compare model parameters and inference efficiency between these methods.
>
> *Table: Additional comparison of model parameter size and inference time.*
>
> | Method | \# Params | Inference Time (ms) |
> |---|---|---|
> | Base | 1.3M | 33 |
> | MMLD | 1.5M | 54 |
> | DRA | 1.5M | 63 |
> | SLDG (Ours) | 1.5M | 68 |
>
>
> **Q4. Broader application of SLDG.**
>
> A4. Our main insight is to iteratively discover latent domains and train personalized classifiers for each latent domain. This idea should be applicable in other scenarios facing similar challenges (i.e., no domain ids and distinct characteristics across domains). Below, we show an additional experiment on the medical imaging classification task. Specifically, we use the MIMIC-CXR data, consisting of chest X-ray data collected between 2008 and 2019. We train a ResNet backbone model on the X-rays before 2014 and test it on X-rays after 2014.
>
> *Table: Additional experiment on chest X-ray classification with temporal domain shift. SLDG can still outperform other DG baselines.*
>
> | Method | ROC-AUC |
> |---|---|
> |Oracle | 0.8011 |
> | Base | 0.7462 |
> | MMLD | 0.7580 |
> | DRA | 0.7446 |
> | SLDG (Ours) | **0.7736** |
>
>
> **Q5. Comparison to federated learning / disentangled representation learning.**
>
> A5. SLDG aims to tackle two main challenges: (1) unknown domain IDs and (2) distinct characteristics across domains. These challenges can be seen in the federated learning setting where different clients have non-iid data. We believe that some of our modules (e.g., decoupled representation learning and latent domain discovery) can also be leveraged in the federated learning setting.
>
> In terms of disentangled representation learning, our method shares similar high-level intuitions. That is, we decouple the patient representation into different latent spaces to discover the view-specific latent domains. In the main experiment, we also include ManyDG, a SOTA disentangled representation learning baseline.
>
>
> [1] Huaxiu Yao et al. Wild-Time: A Benchmark of in-the-Wild Distribution Shift over Time. NeurIPS 2022.
>
> [2] Shiori Sagawa et al. Distributionally robust neural networks for group shifts: On the importance of regularization for worst-case generalization. ICLR 2020.
>
> [3] Bo Li et al. SPARSE MIXTURE-OF-EXPERTS ARE DOMAIN GENER- ALIZABLE LEARNERS. ICLR 2023.
>
> Thank you for taking the time during the rebuttal phase. We hope that these updates address your questions.

---

> > ### Comment · Reviewer_36fo · 2023-08-19
> > **follow up**
> >
> > dear authors,
> >
> > thank you for the repsonse to my questions and the addtional results. i updated my score.
> >
> > in good spirits,
> >
> > reviewer 36fo

---

> > > ### Author Response · Authors · 2023-08-20
> > >
> > > Thank you for acknowledging our response and deciding to raise the score! We really appreciate the valuable feedback and will make sure to include the additional experiments in the final version. Please let us know if you have any further questions. Thanks!

---

### Official Review · Reviewer_RZ7L · 2023-07-10

**Soundness:** 3 good
**Presentation:** 4 excellent
**Contribution:** 3 good
**Rating:** 7
**Confidence:** 3

**Summary:**

The paper presents a novel method for domain generalization in healthcare, named Subspace Latent Domain Generalization (SLDG). The authors argue that traditional domain generalization methods are not suitable for healthcare applications due to the heterogeneity of patient data. The SLDG method discovers latent domains in patient data and uses them to improve the performance of predictive models. The authors provide empirical evidence to support their claims, demonstrating that the SLDG method outperforms several other methods in predicting patient readmission and mortality on two datasets, eICU and MIMIC-IV.

**Strengths:**

* Novel Approach: the paper presents a novel method (SLDG) for domain generalization in healthcare. This method, which discovers latent domains in patient data, could potentially improve the performance of predictive models in healthcare, where data is often heterogeneous and collected from various sources.
* Performance: the proposed method surpassed several other strategies on both readmission and mortality for both datasets used for evaluation.
* Method description: the proposed approach is extensively described with the provided pseudocode. Similarly the methodology employed is properly explained in the paper, with additional material containing the specificities of each section.

**Weaknesses:**

Even though it performed well on the two evaluated datasets, it isn't clear how it would perform on smaller datasets with large amounts of missing data. It is possible the method won't perform so well in a dataset that is smaller than the ones where it was evaluated (1/10th of its size). There is a lack of discussion on the proposed method's limitations and possible challenges within the application.

**Questions:**

* How does the SLDG method handle situations where the latent domains are not clearly separable? Could this affect the performance of the method?
* Was the model evaluated in smaller datasets? What are the scenarios in which this method should or should not be applied?

**Limitations:**

The authors have provided a clear and detailed description of their method and have presented empirical evidence to support their claims. However, they could further discuss the limitations of their method, particularly in terms of the generalizability of the results. The paper focuses on two specific datasets, and it's unclear how well the method would perform on other datasets or in other healthcare contexts. The authors could also discuss the potential challenges of applying the SLDG method in practice, such as the need for large amounts of high-quality data.

---

> ### Author Rebuttal · Authors · 2023-08-10
>
> We would like to thank the reviewer for the positive feedback. We have made the necessary updates to our manuscript. Below we would like to take this opportunity to respond to your questions.
>
>
> **Q1. How would SLDG perform on smaller datasets? Or when the latent domains are not clearly separable?**
>
> A1. It is difficult to provide a general performance conclusion of SLDG on unseen datasets (“No Free Lunch” theory). However, in our model design, we chose an efficient cluster-based parameterization initialization that can better accommodate smaller datasets. In terms of domain separability, we adopt a hierarchical clustering approach based on the Silhouette score which will automatically decide the number of clusters.
>
>
> **Q2. What are the scenarios in which this method should or should not be applied?**
>
> A2. We propose SLDG to tackle two main challenges: (1) unknown domain IDs and (2) distinct characteristics across domains. In the scenario when the domain IDs are given and clearly separable (e.g., photo, art painting, cartoon, and sketch in the PACS dataset), SLDG’s domain discovery approach might be unnecessary. Existing DG methods directly utilizing the domain IDs might be a better solution.
>
> Thank you for taking the time during the rebuttal phase. We hope that these updates address your questions.

---

> > ### Comment · Reviewer_RZ7L · 2023-08-14
> > **After the rebuttal**
> >
> > I thank the authors for their clarifications. I will keep my original score.

---

> > > ### Author Response · Authors · 2023-08-17
> > >
> > > Thanks for your reply! We appreciate you for considering our response and providing positive feedback. We will add more discussion on the limitation of the proposed SLDG method in the final version. Please do not hesitate to let us know if you have any other questions. Thanks!

---

### Official Review · Reviewer_Cuot · 2023-07-27

**Soundness:** 2 fair
**Presentation:** 3 good
**Contribution:** 2 fair
**Rating:** 3
**Confidence:** 4

**Summary:**

This paper introduces a transformer-based supervised learner (SDLG) for two healthcare domain adaptation adaptation tasks (one across geographic region, and one across time).  The pipeline consists of feature sequence embeddings into type-specific vector spaces via fine-tuned contextualized embedding approach, clustered and subclustered, and then passed through a distance-based gating layer.  The model outperform multiple methods from the domain adaptation literature in two electronic health records datasets.

I have read the rebuttal, and while the clarifying comments are elucidating, I maintain my reservations about the work in its current form.

**Strengths:**

Originality: The machine learning modeling approach combines multiple standard models in deep learning and uses appropriate fine-tuning methods.  It is applied to an extract of challenging real world (EHR) data.

Quality: The figures (1,2) and equations effectively communicate the technical approach.

Clarity: The paper is mostly clear and the descriptions are straightforward to follow.

Significance: The results suggest substantial improvement over other baseline models, suggesting that, at least for these tasks, this decoupled approach works well when generalizing to extracts from clinical event sequences in a second domain.

**Weaknesses:**

It seems there could be a slight mismatch in the intuition of having separate latent domains and using the types shown in Fig 2. E.g. toe numbness (symptom), insulin (treatment), and history of diabetes (medical history) are all associated with the condition diabetes, but appear to be encoded separately into spaces/clusters/subclusters.

How UMAP is used is not entirely clear.  Is it performed on the training data only? If it is applied to the training and test data, is it not leaking information from the test in producing the embedding? If not, when applied to the test data, what is the mapping for test data points, and how are they assigned cluster/subcluster memberships?  The "hierarchical clustering" used, if as described, perhaps should be named differently as it does not appear to be the usual hierarchical clustering based on linkage (or details are missing).

The model pipeline includes many design choices, but is tested only in two particular settings.  In those settings some of the details are not entirely clear.  I was also unable to determine the list of event types used, or how the EHR data (typically thousands of measurements) were converted into sequences roughly 100 and 35 events long, respectively.  Which events/tables were used?  How the timestamps are embedded should be described precisely somewhere (e.g. is it time from first event in some unit of time?).

For eICU and given the number of sequential design choices, it seems like leave-out-one-domain cross validation would be a better measure of performance, than a single assessment for the held-out midwest region.  For MIMIC, the Appendix claims each temporal range is split into training/validation/test sets, but also says the evaluation is on patients admitted after 2014, which is confusing.

From an applied perspective, it is unclear what AUCs are competitive or how they would influence decision making.  Are readmission AUCs of 0.62 compelling (or 0.68 for the oracle)?

**Questions:**

See weaknesses above.

**Limitations:**

A high-level set of limitations are described in the Appendix.  While the data used are "real-world", the simplistic extraction used for the event sequence, and experimental setup suggest that the review should likely focus on the methods provided rather than value for the use case.  The methods can still be improved, perhaps with the use of synthetic data where theory and/or optimality could be established.

---

> ### Author Rebuttal · Authors · 2023-08-10
>
> We thank the reviewer for the valuable comments. We would like to take this opportunity to clarify some potential misunderstandings regarding our manuscript.
>
>
> **Q1. Different types of medical codes can be related to each other but are encoded separately.**
>
> A1. We wish to clarify the confusion regarding our feature-specific patient encoding module. Specifically, for a patient’s hospital visit $x = [e_1, . . . , e_m]$, we first pass it through a Transformer encoder to obtain the contextualized event representations, as $[ \mathbf{e}_1, \dots, \mathbf{e}_m ] = E ( [ e_1, \dots, e_m ] )$. The Transformer encoder employs a self-attention mechanism to capture relationships among event pairs. Subsequently, we aggregate the contextualized event representations based on their respective types. This aggregation results in a distinct patient representation that takes into account different medical codes. Consequently, each medical code contributes to the interaction with all other codes within the same patient visit.
>
>
> **Q2. How UMAP is used?**
>
> A2. UMAP and hierarchical domain clustering is restricted to the training set. Specifically, we start by employing the UMAP algorithm to reduce the embedding dimension of each training sample to 2. Subsequently, a hierarchical k-Means algorithm operates in the dimension-reduced latent space, clustering the training samples. During testing, a test sample is mapped to specific clusters based on the distance between its embedding and the mean cluster embeddings.
>
>
> **Q3. Additional experimental details.**
>
> A3. Thank you for the feedback. We will enhance our manuscript to include more implementation and experiment details. We mainly follow existing works [1,2,3,4] in terms of task definition and feature selection. Specifically, for the MIMIC-IV dataset, we use the diagnosis, procedures, and prescriptions events, extracted from “diagnoses_icd”, “procedures_icd”, and “prescriptions” tables; For the eICU dataset, we use the diagnosis, procedures, prescriptions, and lab events extracted from the “diagnosis”, “treatment”, “medication”, “lab” tables. For both datasets, we also include patient demographics information, including age, gender, and ethnicity, extracted from the “patients” and “admissions” tables for the MIMIC-IC dataset, and from the “patient” table for the eICU dataset.
>
> **Q4. The rationale behind the proposed domain split.**
>
> A4. In our original experiment, for both datasets, the target domain is selected as the domain that exhibits the largest performance gap (with a model trained on other domains) in our pilot study. Additionally, we also split each domain into 70%, 10%, and 20% train, val, and test splits. This is to ensure that the oracle, base, and DG methods are comparable. Specifically, the oracle is trained on the target training and validation set, and tested on the target testing set. In contrast, the base and DG methods are trained on the source training and validation set and then tested on the target testing set.
>
> **Q5. Are readmission AUCs of 0.62 compelling?**
>
> A5. The readmission prediction task is very critical in real-world clinical practice to assist hospital planning and improve patient outcomes. This task is studied in several existing works [4,5,6] and grants (NIH 5K23HL153582-03, NIH 1R01HL119664-01A1, NIH 1R01HL130828-01A1). In our study, we adopt a stricter setup following Guo et al. [7]. For example, for the eICU dataset, the readmission prediction is made 12 hours after ICU admission (compared to the end to current admission in ManyDG [6]). It is important to note that we use the same setup across all methods for a fair comparison. The performance gap between the baseline and oracle method should be the most important indicator of model effectiveness.
>
>
> [1] Edward Choi et al. RETAIN: An Interpretable Predictive Model for Healthcare using Reverse Time Attention Mechanism. NIPS 2016.
>
> [2] Edward Choi et al. Using recurrent neural network models for early detection of heart failure onset. JAMIA 2017.
>
> [3] Phuoc Nguyen et al. Deepr: A Convolutional Net for Medical Records. IEEE JOURNAL OF BIOMEDICAL AND HEALTH INFORMATICS 2017.
>
> [4] Chaohe Zhang et al. GRASP: Generic Framework for Health Status Representation Learning Based on Incorporating Knowledge from Similar Patients. AAAI 2021.
>
> [5] Rich Caruana et al. Intelligible Models for HealthCare: Predicting Pneumonia Risk and Hospital 30-day Readmission. KDD 2015.
>
> [6] Chaoqi Yang et al. MANYDG: MANY-DOMAIN GENERALIZATION FOR HEALTHCARE APPLICATIONS. ICLR 2023.
>
> [7] Lin Lawrence Guo et al. Evaluation of domain generalization and adaptation on improving model robustness to temporal dataset shift in clinical medicine. Scientific Reports 2022.
>
>
> Thanks for your time during the rebuttal phase. We hope the updates can clarify the misunderstandings and address the reviewer’s concerns.

---

> > ### Author Response · Authors · 2023-08-20
> > **We would like to hear back from reviewer Cuot**
> >
> > Dear reviewer Cuot,
> >
> > We would like to follow up to see if the response addresses your concerns or if you have any further questions. We would really appreciate the opportunity to discuss this further if our response has not already addressed your concerns. Thank you again!
> >
> > Submission 9784 Authors

---

> ### Comment · Area_Chair_CAX4 · 2023-08-21
> **Please check the author response**
>
> Dear Reviewer,
>
> I would like to know if the author rebuttal has addressed your concerns. Specifically, please acknowledge that you've read the review, update your score to reflect a change in assessment, if any, and let me know if you still have concerns about the paper. The authors work very hard during the rebuttal phase and it is important to ensure that all input is accounted for to make a final decision.
>
> Thank you!
> AC

---

> > ### Comment · Reviewer_Cuot · 2023-08-21
> >
> > The original summary paragraph was edited to indicate that I have reviewed the response and maintained the rating. More specifically, the embedding approach and the evaluation setup have been clarified, and how UMAP is used with respect to the test data is clearer.  Yet the central concerns remain regarding the usability of the approach in the applied domain, and therefore one might expect the methodological contribution to be larger.  For example, for readmission prediction, why (from a use standpoint) would one use up to 12 hours past ICU admission for this task?  There are tables in eICU discussing "Care Plans" which one would likely want to consider for use in readmission prediction, so a use-case justification for table inclusion is warranted (likewise for the time cut-off, table row inclusion).  As another example, what motivates the use of labs in eICU but not MIMIC?  Is it the regularity of the coding of these tables an issue, and what are the ramifications with respect to OOD evaluation?  Is it the event sequence length change that impairs performance?  We would want a new OOD method to work robustly to some of these design choices, and hence the question about evaluating on the midwest region (as opposed to CV), and under some larger set of reasonable design choices.

---

> > > ### Author Response · Authors · 2023-08-21
> > >
> > > Thank you for the comments.
> > >
> > > We would like to emphasize that the main focus of our study is to develop a clinical predictive model that is generalizable against potential domain shifts. This distinguishes our work from the sole objective of achieving high absolute scores, which has been extensively studied in previous works [1-10]. While we understand your concerns about the specifics and rationale behind our experimental setup, we believe these concerns can be appropriately handled based on various real-world requirements. Therefore, these concerns should not diminish the practical significance of the problem or the novelty of the proposed method.
> > >
> > > ---
> > >
> > > **Q1. Why was data only utilized up to 12 hours after ICU admission?**
> > >
> > > A1. The mortality and readmission prediction tasks have received considerable attention in the deep learning healthcare domain [1-10]. The prevailing setup involves predicting future events (mortality/readmission) based on preceding events occurred within an observation window. For instance, Yang et al. [6] employed data spanning the entire admission period for readmission prediction, whereas Li et al. [1] considered data up to 24 hours post-ICU admission. Our study adopted a stringent setup that exclusively incorporates data within the initial 12 hours following ICU admission. We opted for this approach to filter out fewer admissions of shorter duration and to enable earlier clinical intervention in real-world practice. We must note that we use the same setup across all methods for a fair comparison. The performance gap between the baseline and oracle method should be the most important indicator of model effectiveness.
> > >
> > > **Q2. Why weren't the "Care Plans" / "Labs" tables from eICU utilized?**
> > >
> > > A2. Our feature selection closely aligns with existing studies [6] to facilitate a direct comparison. In fact, a substantial amount of information in the "Care Plans" table can also be found in the other selected tables (e.g., admission diagnosis, infection). Nonetheless, our method can readily adapt to different setups based on real-world necessities. We believe this should not undermine the problem's practical significance or the proposed method's novelty.
> > >
> > > **Q3. How robust is the proposed method against different settings or design choices?**
> > >
> > > A3. In our initial experiments, we reported average scores and standard deviations through 1000 bootstrapping iterations. Moreover, we employed independent two-sample t-tests to gauge whether SLDG exhibited significant enhancements over baseline methods. We also analyzed the impact of two important hyper-parameters: the number of clusters and iterations. During the rebuttal phase, we provided additional experiments to show the broader application of SLDG on the medical imaging classification task. We welcome any specific suggestions from the reviewer regarding additional evaluations/experiments.
> > >
> > > *Table: Additional experiment on the chest X-ray classification task with temporal domain shift. Specifically, we used the MIMIC-CXR data, consisting of chest X-ray data collected between 2008 and 2019. We trained a ResNet backbone model on the X-rays before 2014 and tested it on X-rays after 2014. Results show that SLDG can still outperform other DG baselines.*
> > >
> > > | Method | ROC-AUC |
> > > |---|---|
> > > |Oracle | 0.8011 |
> > > | Base | 0.7462 |
> > > | MMLD | 0.7580 |
> > > | DRA | 0.7446 |
> > > | SLDG | 0.7736 |
> > >
> > > ---
> > >
> > > Thank you once again for your time. We hope the response can address your concerns and clarify the contributions of our work.
> > >
> > > ---
> > >
> > > [1] Edward Choi et al. RETAIN: An Interpretable Predictive Model for Healthcare using Reverse Time Attention Mechanism. NIPS 2016.
> > >
> > > [2] Edward Choi et al. Using recurrent neural network models for early detection of heart failure onset. JAMIA 2017.
> > >
> > > [3] Phuoc Nguyen et al. Deepr: A Convolutional Net for Medical Records. IEEE JOURNAL OF BIOMEDICAL AND HEALTH INFORMATICS 2017.
> > >
> > > [4] Chaohe Zhang et al. GRASP: Generic Framework for Health Status Representation Learning Based on Incorporating Knowledge from Similar Patients. AAAI 2021.
> > >
> > > [5] Rich Caruana et al. Intelligible Models for HealthCare: Predicting Pneumonia Risk and Hospital 30-day Readmission. KDD 2015.
> > >
> > > [6] Chaoqi Yang et al. MANYDG: MANY-DOMAIN GENERALIZATION FOR HEALTHCARE APPLICATIONS. ICLR 2023.
> > >
> > > [7] Lin Lawrence Guo et al. Evaluation of domain generalization and adaptation on improving model robustness to temporal dataset shift in clinical medicine. Scientific Reports 2022.
> > >
> > > [8] Hrayr Harutyunyan et al. Multitask learning and benchmarking with clinical time series data. Scientific Data 2019.
> > >
> > > [9] Min Hyuk Choi et al. Mortality prediction of patients in intensive care units using machine learning algorithms based on electronic health records. Scientific Reports 2022.
> > >
> > > [10] Yikuan Li et al. BEHRT: Transformer for Electronic Health Records. Scientific Reports 2020.

---

### Official Review · Reviewer_cG9z · 2023-07-28

**Soundness:** 3 good
**Presentation:** 2 fair
**Contribution:** 2 fair
**Rating:** 6
**Confidence:** 3

**Summary:**

* This paper proposes a learning framework for improved predictive performance for EHR data under distribution shift.
* The key problem tackled here is that many existing domain generalization algorithms rely on data points having domain identifiers, which can then be used in a learning algorithm to build a generalizable model. However, in clinical settings, patients may fall into many domains, which may also be latent (and not explicitly known), making the application of these existing methods challenging.
* In this work, the proposed framework iterates (1) domain discovery and (2) domain-specific classifier training, to build predictors that are effective for the different domains considered.
* This framework involves:  obtaining encoded representations for patient visits, clustering visits using a hierarchical procedure to discover domains, and building a weighted nearest neighbour styled classifier within each domain (using prototypes) to predict a label.
* The paper evaluates the method and several baselines on two EHR datasets: eICU and MIMIC-IV. Two tasks are considered: readmission, and mortality prediction. On both datasets, the method performs well, and improves on baselines.
* The paper also includes ablations on the number of iterations of the procedure and clusters, providing some more intuition on what drives performance.


**Strengths:**

* Clear presentation of the problem and limitations of existing methods e.g., the need for domain IDs, or training a single model across domains.
* As far as I can tell, method appears to be original and also well motivated based on the problem at hand -- discovering latent regions of similarity and developing specialized classifiers for those regions.
* Good evaluation setup: fairly comprehensive set of baselines, evaluation on two distinct datasets with varied characteristics
* Results are encouraging (though a question/comment on this below) -- consistent improvements in AUROC/AUPRC on the two tasks on both datasets.
* Useful additional results studying importance of clustering algorithm and understanding the nature of the latent spaces.

**Weaknesses:**

* Overall I was confused by the methodological presentation. Particularly, a diagram presenting the data analogously to the abstraction  (medical record is a set of visits, each with events) would help a lot. The dimensionality of different data/features at points in the pipeline was unclear, and a figure might help this.
* Similar to the above, Figure 2 showing the method was useful, but I think too high level to properly internalize what was happening in the method. An architecture diagram for the encoder (for example) could help, and may address the point above too.
* In general, Section 3.1 was tricky to understand -- perhaps this is because the high level presentation at the start was difficult to map to the more specific descriptions later (namely, the Feature specific patient encoding and hierarchical domain clustering paragraphs). Some of this confusion might stem from not understanding the input data very well, particularly the different feature types $t$ and encoder $E_t$, and embedding function $E$. A re-writing of this section might help, together with clear figures for each of the pieces (particularly the encoding section).
* On the quantitative results: I appreciate that the proposed method improves on the different baselines -- that is encouraging to see. However, the fact that it does not improve on the Oracle method (and in some cases does substantially worse) may be something to discuss further -- why would we not just use the oracle if we have enough data from that domain?  Would finetuning the proposed method on the training set from the target domain help here (for some definition of finetuning? I appreciate it's a bit tricky here given the way the classifiers are parameterized)? In addition, could you show results when you have very limited data from the target domain (and specifically demonstrate that the Oracle no longer performs well)? As I see it, if the Oracle method that is trained on the labelled target data can do well in small target data regimes, then these results are not as convincing.
* On evaluation of clustering results: Since the proposed method directly looks at the Silhouette score when doing clustering (and not sure the others do), I'm not sure this is very convincing -- we'd expect the proposed method to do better when evaluating Silhouette score.

**Questions:**

* Could you say more about the specific dimensionality and form of the input sequence that the model sees? And as above, adding a clearer figure to represent the data and the notational abstraction used in the paper. Adding figure to specify the architecture, and how the encoding phase works in more detail.
* Can you answer my question re. the oracle, as discussed in Weaknesses? Specifically, on the proposed method being (sometimes substantially) worse than the oracle, and how to think about this?

Overall, if the authors can say more about the above two points, I will increase my score.


# After rebuttal
Authors have answered my main questions, so I will increase my score.

**Limitations:**

* Limitations and broader impact are discussed in the appendix. I think however that the limitations could be fleshed out some more, maybe in terms of acknowledging that only two predictive tasks and two datasets are considered.

---

> ### Author Rebuttal · Authors · 2023-08-10
>
> We thank the reviewer for the valuable feedback. We would like to take this opportunity to clarify some potential misunderstandings regarding our manuscript.
>
>
> **Q1. More explanation on the input data structure and model architecture.**
>
> A1. In our setting, each input sample is defined as a patient’s hospital/ICU visit, which can be represented as a sequence of clinical events. For example, a patient visit can be [Acute embolism (I82. 40), Atrial fibrillation (I48.91), Ultrasound (76700), CT scan (G0296), ECG (93042), Heparin IV (5224)]. In our dataset, each event corresponds to a specific type (e.g., diagnosis, procedure, medication) and a specific timestamp (i.e., when the event occurred). Additionally, we add special <cls> and <eos> tokens at the beginning and end of the event sequence. To embed the patient visit, we use a three-layer Transformer encoder which essentially takes each medical code as a token and performs the self-attention mechanism across different codes.
>
> Please see the attached PDF file for a detailed illustration. We will also incorporate some examples in the manuscript for better understanding.
>
>
> **Q2. Why is the proposed method weaker than the Oracle method?**
>
> A2. We would like to clarify that the Oracle method is directly trained on the target domain, while the rest of the baselines (base and DG methods) are only trained on the source domain and later tested on the unseen target domain. As all methods use the same backbone encoder, the Oracle method can be viewed as an upper bound of the model performance. As a result, the goal of all DG methods is to match the performance of the in-domain-trained oracle without seeing any in-domain data. And the performance gap between Oracle and DG methods should be the main indicator of model effectiveness.
>
> In real-world clinical practice, due to privacy and legal concern, it is difficult to obtain the target data (e.g., from other hospitals) to train the Oracle method. Thus, DG solutions that can match the performance of the oracle but without the need of accessing target data are preferred.
>
> **Q3. Evaluation of clustering on the Silhouette score.**
>
> A3. We would like to clarify that during the clustering step, the hyper-parameters are chosen based on the Silhouette score on the training set; while during the evaluation, the Silhouette score is calculated on the testing set. Further, the model is later fine-tuned with the domain-specific classifiers. So there is no unfair advantage of SLDG. Nevertheless, we report the Calinski-Harabasz score (the higher, the better) of SLDG and the best baseline MMLD below.
>
> *Table: Results with additional cluster metric (Calinski-Harabasz score, the higher, the better).*
>
> | Method | eICU readmission | eICU mortality | MIMIC-IV readmission | MIMIC-IV readmission |
> |---|---|---|---|---|
> | MMLD | 137.9  | 125.8 | 169.8 | 264.7 |
> | SLDG (Ours) | **246.5** | **302.5** | **233.1** | **355.9** |
>
>
> Thanks for your time during the rebuttal phase. We hope the updates can clarify the misunderstandings and address the reviewer’s concerns.

---

> > ### Comment · Reviewer_cG9z · 2023-08-16
> > **Thank you for your response!**
> >
> > I thank the authors for their response and for addressing some of the misunderstandings. I will raise my score accordingly.

---

> > > ### Author Response · Authors · 2023-08-17
> > >
> > > Thank you for taking the time during the rebuttal phase. We are grateful to you for acknowledging our response and deciding to raise the score. In the final version, we will include more discussion on the input data, model architecture, Oracle method, and the use of the Silhouette score. Please do not hesitate to let us know if you have any other questions. Thanks!

---

### Author Rebuttal · Authors · 2023-08-10

Additional PDF illustrating the input data structure, task definitions, and encoder architecture.

---

### Author Response · Authors · 2023-08-21
**Summary of the Rebuttal Phase**

We sincerely thank all reviewers for investing their time during the rebuttal phase. Their constructive comments have been invaluable to the enhancement of our manuscript.

As a summary of the rebuttal phase, we have made several revisions to our manuscript, including the following new results:

1. additional metric on the clustering result,
2. additional comparison with recent domain generalization methods,
3. detailed analysis of the computational complexity,
4. broader application of SLDG on the chest X-ray classification task.

We have also added detailed explanations of the model pipeline and experiment setup, and expanded discussions of related works and the limitation of SLDG.

We believe these revisions have enriched our manuscript. We hope our revised manuscript will meet the high standards of this conference and contribute meaningfully to the field of machine learning for healthcare.

---

### Decision · Program_Chairs · 2023-09-21

**Decision:**

Accept (poster)

**Comment:**

This paper proposes a domain generalization framework relevant to healthcare. The key idea is that "domains" in health are not as clear as other applications. Thus the authors provide an iterative framework that learn the domains using an embedding+hierarchical clustering. The domain generalization framework + latent domain identification together forms their self-learning framework. Authors demonstrate experiments on MIMIC and eICU, standard health benchmarks.

Reviewers overall have not found major technical issues, although several clarity issues have been identified. Authors have addressed major concerns from the reviewers including additional baselines, which I think should have been part of the original contribution (such as GroupDRO). My own view of the paper is that UMAP and related methods are fairly unreliable but might be acceptable for a downstream classification task. As such I expect the authors to include limitations of their unsupervised part of the framework.

Besides that I would like to encourage authors to be more precise for clarity and include all additional empirical evaluation in the final version. Assuming the authors will deliver on this, I recommend an accept.